# Growth, Flowering, and Fruit Production of Strawberry ‘Albion’ in Response to Photoperiod and Photosynthetic Photon Flux Density of Sole-Source Lighting

**DOI:** 10.3390/plants12040731

**Published:** 2023-02-07

**Authors:** Yujin Park, Rashmi Sethi, Stephanie Temnyk

**Affiliations:** College of Integrative Sciences and Arts, Arizona State University, Mesa, AZ 85212, USA

**Keywords:** controlled environment, fruit production, hydroponics, light intensity, photoperiod, sole-source LED lighting, strawberry, vertical farming

## Abstract

Beyond producing leafy greens, there is a growing interest in strawberry production on indoor vertical farms. Considering that sole-source lighting is one of the most important components for successful indoor crop production, we investigated how photosynthetic photon flux density (PPFD) and the photoperiod of sole-source lighting affected plant growth, flowering, and fruit production in strawberry ‘Albion.’ Bare-rooted strawberry plants were grown in deep water culture hydroponics inside an indoor vertical farm at 21 °C under white + blue + red light-emitting diodes at a PPFD of 200, 300, or 450 µmol∙m^−2^∙s^−1^ with a 12-h or 16-h photoperiod. Under both photoperiods, increasing PPFD from 200 to 450 µmol∙m^−2^∙s^−1^ linearly increased crown diameter by 18–64%, shoot fresh and dry mass by 38–80%, and root fresh and dry mass by 19–48%. Under a PPFD ≥ 300 µmol∙m^−2^∙s^−1^, root fresh and dry biomass increased by 95–108% and 41–44%, respectively, with an increasing photoperiod from 12 to 16 h. In addition, increasing the photoperiod from 12 to 16 h accelerated flowering by 17–21 days under a PPFD ≥ 300 µmol∙m^−2^∙s^−1^ and first fruit harvest by 17 days at a PPFD of 450 µmol∙m^−2^∙s^−1^. Regardless of PPFD, strawberry fruit production (g·m^−2^·month^−1^) increased by 372–989% under a 16-h photoperiod in comparison to under a 12-h photoperiod. In contrast, there was little effect of PPFD on fruit production. Our results suggest that increasing the PPFD or photoperiod can increase strawberry plant growth, but increasing the photoperiod can have a dominant effect on increasing early fruit production in strawberry ‘Albion’.

## 1. Introduction

There is an increasing interest in growing strawberries (*Fragaria* × *ananassa*) via indoor vertical farming techniques. Though leafy greens dominate the vertical farming landscape for their established reliability in productivity, major players in the U.S. berry industry have invested millions of dollars in growing strawberries indoors, including Manhattan-based startup Oishii and vertical farming company Plenty paired with Driscoll’s [1]. The impetus is enticing: indoor strawberry production boasts reduced resource utilization coupled with increased productivity, proximity to local markets both domestic and overseas, and the elimination of weather fluctuation concerns [2]. These boosts in production efficiency will be much needed considering the increased consumer demand for strawberries, with shipments to the U.S. increasing 4% year-over-year in 2020 [1]. Nevertheless, it is still challenging to produce strawberries indoors due to the fruit’s long production cycle, complicated crop development, and intricate environmental interactions compared to leafy greens [1,3]. Such challenges necessitate research to determine the optimal growing conditions for indoor strawberry production.

Artificial lighting is one of the major environmental factors affecting crop growth, yield, and productivity. In indoor vertical farms, light-emitting diodes (LEDs) are commonly used as a sole-source lighting to deliver the light needed for plant growth and development [4,5]. To provide desirable light quantity via sole-source lighting, many previous studies have examined the effect of photosynthetic photon flux density (PPFD), photoperiod, and daily light integral (DLI) of sole-source lighting on plant growth. Though most research has been conducted on leafy vegetables, increasing DLI by increasing the PPFD or photoperiod generally increases crop yield. For example, in basil (*Ocimum basilicum* L.) and lettuce (*Lactuca sativa* L.), when the photoperiod was kept constant at 16 h and PPFD was gradually increased from 100 µmol∙m^−2^∙s^−1^ to 300 µmol∙m^−2^∙s^−1^, a steady increase in biomass production was seen up to 250 µmol∙m^−2^∙s^−1^ [6]. In lettuce ‘Rex’ and ‘Rouxai’, increasing DLI from 6.9 to 15.6 mol·m^−2^·d^−1^ increased fresh and dry mass regardless of different combinations of PPFD and photoperiod [7]. However, at higher DLIs, a DLI given with longer photoperiods and lower PPFDs increased plant growth. For example, at a DLI of 15.6 mol·m^−2^·d^−1^, lettuce ‘Rex’ and ‘Rouxai’ grown with a 24-h photoperiod had 11–27% greater fresh and dry mass than those grown with a 16- or 20-h photoperiod [7]. Similarly, at a DLI of 16 mol·m^−2^·d^−1^, as the photoperiod increased from 10 to 20 h, mizuna (*Brassica rapa* var. japonica) and lettuce ‘Little Gem’ showed 16–19% higher shoot dry mass with increases in leaf chlorophyll content index, canopy light interception, and quantum yield of photosystem II [8].

Recently, a few studies also investigated the effects of PPFD, photoperiod, and DLI of sole-source lighting on plant growth in strawberry plants. When strawberry ‘Albion’ seedlings were grown as stock plants at a PPFD of 250, 350, and 450 µmol∙m^−2^∙s^−1^ with a 12-h photoperiod, the number of daughter plants per stock plant and the crown diameter, leaf number, leaf area, and shoot fresh and dry mass of stock plants increased linearly with increasing PPFD [9]. In strawberry ‘Benihoppe’, increasing PPFD from 90 to 270 µmol∙m^−2^∙s^−1^ at a 16-h photoperiod (or DLI from 5.2 to 15.6 mol·m^−2^·d^−1^) increased the shoot and root dry mass of the runner plants by 39% and 106%, respectively [10]. In strawberries ‘Elan’ and ‘Yotsuboshi’, seedlings grown with a 16- and 24-h photoperiod had a 27–128% higher net assimilation rate and 34–142% greater dry mass at the transplant stage than those with 8- and 12-h photoperiods under the same DLI of 10 mol·m^−2^·d^−1^ [11]. These results suggest that although the recommended DLIs are higher for fruiting crops than for leafy greens and herbs [12,13], increasing DLI can increase the plant growth of strawberries as in leafy greens, and extending photoperiod with a lower PPFD can further increase plant growth at relatively higher DLI levels.

However, previous studies [9,10,11] that have investigated the effects of PPFD, photoperiod, and DLI of sole-source lighting on indoor strawberry production have been limited to the vegetative growth and propagation stage and have not investigated the flowering and fruit yield responses. In the present study, we investigated how PPFD, photoperiod, and DLI influence plant growth, flowering, and fruit yield in everbearing strawberry ‘Albion’ in an indoor vertical farm with sole-source lighting. The flowering of many strawberry cultivars depends on photoperiod [14]. A longer photoperiod accelerated flowering responses in everbearing strawberry cultivars, such as ‘Albion’, ‘Monterey’, and ‘San Andreas’ [14]. Thus, in strawberry cultivation using everbearing cultivars, a longer photoperiod can promote plant growth, as well as flowering, potentially increasing fruit yield. We hypothesized that (1) increasing DLI either by increasing PPFD or increasing the photoperiod would increase plant growth and fruit yield; and (2) an increase in the photoperiod would have a greater effect than an increase in PPFD on growth, flowering, and fruit development in an everbearing strawberry cultivar.

## 2. Results

### 2.1. Plant Growth

Increasing PPFD from 200 to 450 µmol∙m^−2^∙s^−1^ increased crown diameter linearly by 18% under a 12-h photoperiod and 64% under a 16-h photoperiod (Figure 1). The effects of PPFD on leaf length and SPAD index depended on the photoperiod. Leaf length decreased linearly by 19% with an increasing PPFD only under a 16-h photoperiod. Increasing the PPFD increased the SPAD index linearly under a 12-h photoperiod, but not under a 16-h photoperiod. PPFD did not affect the number of leaves, plant diameter, and leaf width. Extending the photoperiod alone from 12 h to 16 h did not affect the number of leaves, crown diameter, plant diameter, leaf size, and SPAD index in strawberry ‘Albion’.

Increasing the PPFD from 200 to 450 µmol∙m^−2^∙s^−1^ linearly increased shoot fresh and dry mass by 53% and 80%, respectively, under a 12-h photoperiod and shoot dry mass by 38% under a 16-h photoperiod (Figure 2 and Figure 3). Increasing the PPFD from 200 to 450 µmol∙m^−2^∙s^−1^ linearly increased root fresh mass by 31% under a 12-h photoperiod and 48% under a 16-h photoperiod, and root dry mass by 19% under a 16-h photoperiod. PPFD, root fresh, and dry mass were 95–108% and 34–44% greater under a 16-h photoperiod, respectively, compared to under a 12-h photoperiod except for the root fresh mass at a PPFD of 200 µmol∙m^−2^∙s^−1^. In contrast, the photoperiod had little effect on shoot fresh and dry mass.

### 2.2. Flowering and Fruit Production

PPFD had little effect on days to flowering, days to first fruit harvest, flowering %, and fruiting % (Figure 4). In contrast, increasing the photoperiod from 12 h to 16 h accelerated flowering by 17–21 days under a PPFD ≥ 300 µmol∙m^−2^∙s^−1^ and first fruit harvest by 17 days under a PPFD of 450 µmol∙m^−2^∙s^−1^. In addition, at a PPFD of 200 and 450 µmol∙m^−2^∙s^−1^, 43–47% more strawberry plants reached flowering under a 16-h photoperiod compared to a 12-h photoperiod. At each PPFD, 50–70% more plants developed fruits under a 16-h photoperiod than a 12-h photoperiod.

There were no significant effects of PPFD and photoperiod on the fresh mass and diameter of individual fruit (Figure 5). PPFD also did not influence the total fruit production and total number of fruits. However, at each PPFD, the total number of fruits was 365–1220% greater under a 16-h photoperiod than under a 12-h photoperiod. With little effects of photoperiod on individual fruit fresh mass, total fruit fresh mass showed a similar pattern as total number of fruits in relation to the photoperiod. Under a 16-h photoperiod compared to a 12-h photoperiod, total strawberry fruit production (g·m^−2^·month^−1^) increased by 372–989%.

## 3. Discussion

### 3.1. Effects of PPFD on Vegetative Growth, Flowering, and Fruit Yield in Strawberries

Increasing PPFD can increase plant photosynthesis and biomass accumulation up to certain PPFD levels, although the saturating PPFD levels depend on plant species, as well as other environmental conditions [15]. For example, in lettuce and basil, as PPFD increases from 100 to 250 µmol∙m^−2^∙s^−1^, shoot fresh and dry mass increased by 104–138% and 175–208%, respectively, whereas they did not change in basil and decreased in lettuce under a further increase of PPFD up to 300 µmol∙m^−2^∙s^−1^ [6]. In cherry tomato (*Lycopersicon esculentum* Mill) seedlings, increasing the PPFD in the range of 50–300 µmol∙m^−2^∙s^−1^ increased fresh and dry mass by 96% and 195%, respectively, with little to no changes in plant biomass at a PPFD of ≥300 µmol∙m^−2^∙s^−1^ [16]. In this study, we observed that increasing the PPFD up to 450 µmol∙m^−2^∙s^−1^ generally increased shoot and root biomass in strawberry ‘Albion’ plants during their vegetative growth without showing a saturating response (Figure 2). Similarly, the plant biomass of strawberry stock plants increased up to a PPFD of 400 µmol∙m^−2^∙s^−1^ in ‘Mae-hyang’ [17] and a PPFD of 450 µmol∙m^−2^∙s^−1^ in ‘Albion’ [9] without saturating responses. In strawberry ‘Marmolada’ and ‘Darselect’, the light saturation point for leaf photosynthesis was observed at a PPFD of around 1400 µmol∙m^−2^∙s^−1^ [18]. These results suggest that the point at which saturation occurs for promoting plant biomass accumulation in strawberry plants can be higher than at a PPFD of 400–450 µmol∙m^−2^∙s^−1^.

The strawberry crown (a shortened thick stem) is an important plant growth parameter that affects strawberry plant vigor and fruit yield since it acts as a crucial source of carbohydrates for fruit growth [19,20,21]. Bare-rooted transplants with crown diameters above 10 mm enhanced early and total fruit production in three strawberry cultivars [21]. In this study, when the bare-rooted plants with a crown diameter of 10–11 mm were grown for five weeks under sole-source lighting, the crown diameter linearly increased by 18–64% when increasing PPFD from 200 to 450 µmol∙m^−2^∙s^−1^ (Figure 1). This result is consistent with other studies on strawberries. Increasing the PPFD from 90 to 360 µmol∙m^−2^∙s^−1^ increased the crown diameter by 15% of strawberry ‘Benihoppe’ rooted runner plants [10]. Strawberry ‘Toyonoka’ stock plants had 41–43% greater crown diameter under a PPFD of 110–122 µmol∙m^−2^∙s^−1^ compared to under a PPFD of 50–55 µmol∙m^−2^∙s^−1^ [22]. As PPFD increased from 250 to 450 µmol∙m^−2^∙s^−1^, the crown diameter of strawberry ‘Albion’ stock plants linearly increased by 40% [9]. In addition, the increases in crown diameter with increasing PPFD in this study were accompanied by the increases in shoot biomass (Figure 1 and Figure 2). Given that strawberry plants with greater crown diameter had greater carbohydrate content in the crown [23,24], increasing the PPFD may have contributed to increased crown diameter by promoting carbohydrate accumulation in the crown.

As a rule of thumb, a 1% increase in light quantity is assumed to lead to a 1% increase in yield. In fruit vegetable crops, such as cucumber and tomato, an average increase of 0.7–1% in harvestable yield is expected with 1% additional light [25]. In this study, increasing the PPFD from 200 to 450 µmol∙m^−2^∙s^−1^ had little to no significant effect on fruit production in strawberry ‘Albion’ at both photoperiods (Figure 5). The effect of light quantity on crop yield can be limited when other environmental and cultural factors are the major limiting factors [15,25]. In strawberries, fruit yield can depend on multiple environmental factors, such as light, temperature, humidity, nutrition, irrigation, and CO_2_; and growth factors, such as crown size, leaf size, and flowering [21,26,27,28]. In this study, strawberry fruit production showed a similar pattern with flowering and fruiting percentage, which was mainly governed by the photoperiod. A 12-h photoperiod suppressed and a 16-h photoperiod promoted flowering, fruit development, and subsequent fruit production, regardless of PPFD. The minor effects of PPFD seen on fruit yield in our study are, therefore, likely from the dominating effects of the photoperiodic flowering response.

### 3.2. Effects of Photoperiod on Vegetative Growth, Flowering, and Fruit Yield in Strawberries

Similar to the effects of increasing the PPFD at a given photoperiod, increasing the photoperiod at a given PPFD increases the DLI and, typically, the plant biomass. For example, increasing the photoperiod from 12 to 16 h at PPFDs of 100–300 µmol∙m^−2^∙s^−1^ increased the fresh and dry mass of shoot and root of cucumber (*Cucumis sativus*) ‘Jintong’ and ‘Yunv’ seedlings by 24–185% [29]. In lettuce, the fresh and dry mass of leaves and roots generally increased by 5–91% when increasing the photoperiod from 12 to 16 h at PPFDs of 150–300 µmol∙m^−2^∙s^−1^ [30]. In this study, increasing the photoperiod from 12 to 16 h increased the DLI by 33% and the root dry mass of strawberry ‘Albion’ plants by 34–44% (Figure 2). Considering that increasing the DLI by 125% through increasing the PPFD from 200 to 450 µmol∙m^−2^∙s^−1^ had little effect on root dry mass under a 12-h photoperiod, or increased root dry mass by only 19% under a 16-h photoperiod, increasing the DLI by increasing the photoperiod had greater promotive effects on root biomass accumulation than by increasing PPFD. In contrast, increasing the photoperiod had little effect on shoot biomass of strawberry ‘Albion’ in this study.

Photoperiodic flowering responses of strawberry plants are often classified into three categories, including June-bearing (short-day), everbearing (long-day), and day-neutral types [31]. However, the classification of photoperiodic flowering responses has been less clear in some modern strawberry cultivars developed in California, such as ‘Albion’ [32], ‘Monterey’ [33], ‘Portola’ [34], and ‘San Andreas’ [35], as their flowering types are described as everbearing and day-neutral at the same time. This is at least partly because their photoperiodic flowering response depends on temperature. Everbearing strawberry cultivars tend to show day-neutral responses at low temperatures (below 10 °C), whereas they are quantitative long-day plants at intermediate temperatures (between 10 and 27 °C) and qualitative long-day plants at high temperatures (27 °C) [36,37]. A recent study clearly showed a facultative long-day response in strawberry ‘Albion’, ‘Monterey’, and ‘San Andreas’ at 17 °C, as increasing the photoperiod from 8 to 17 h accelerated floral meristem development [14]. Similarly, in our study at 21 °C, as the photoperiod increased from 12 to 16 h, strawberry ‘Albion’ flowered 17–21 days faster under a PPFD of ≥300 µmol∙m^−2^∙s^−1^ (Figure 4), exhibiting a quantitative long-day response in flowering. 

However, increasing the photoperiod did not promote the flowering of strawberry ‘Albion’ at a PPFD of 200 µmol∙m^−2^∙s^−1^ (Figure 4). Strawberry plants use carbohydrate reserves accumulated in the crown and roots to support initial flower and fruit development [24,38]. Ito and Saito [39] reported that the size and age of strawberry plants affect their sensitivity to photoperiod and temperature, with larger and older plants having greater sensitivity than smaller and younger ones. In this study, increasing the PPFD increased the crown diameter and plant biomass (Figure 1 and Figure 2), and thus, a smaller plant size with a smaller crown diameter under a PPFD of 200 µmol∙m^−2^∙s^−1^ could have contributed to a less sensitive photoperiodic flowering response.

It is recommended to maintain a minimum and optimum DLI of 10–12 mol∙m^−2^∙d^−1^ and 20–25 mol∙m^−2^∙d^−1^, respectively, for good strawberry production in greenhouses [40]. However, even within the desirable DLI range, our results showed that when similar DLIs were delivered with a 12-h photoperiod, strawberry ‘Albion’ plants produced much fewer fruits compared to under a 16-h photoperiod (Figure 5). In strawberries, controlling flowering is the key to early fruit production, as early flower initiation leads to earlier fruiting [41]. In this study, earlier flowering (by 17–21 days) in strawberry ‘Albion’ under a 16-h photoperiod compared to a 12-h photoperiod led to increases in the percentage of plants that produce flowers (by 43–47%) and fruits (by 50–70%) and, subsequently, the increases in total fruit production (by 372–989%) (Figure 4 and Figure 5). In contrast, under a 12-h photoperiod, delayed flowering led to delayed fruiting, significantly decreasing the fruiting percentage and early fruit production. Our results suggest that delivering DLIs with an inductive long photoperiod can enhance the initial productivity of strawberry via early flowering and fruiting in at least some everbearing strawberry cultivars.

## 4. Materials and Methods

### 4.1. Plant Materials

Bare-rooted strawberry ‘Albion’ plants were obtained from a commercial nursery (Lassen Canyon Nursery, Redding, CA, USA) on 3 September 2020 (replication 1) and 14 January 2021 (replication 2). Upon the arrival of the bare-rooted strawberry plants on the Arizona State University (ASU) Polytechnic campus (Mesa, AZ, USA), they were placed into a commercial cooler at 0 °C until transplanting on 17 September 2020 (replication 1) and 15 February 2021 (replication 2). In each replication, 180 uniform bare-rooted strawberry plants were selected, based on their crown diameter of 10–11 mm, for transplanting. Each bare-rooted strawberry plant was washed with tap water to remove any substrate on the roots. To prevent plant pathogens, the cleaned bare-rooted strawberry plants were soaked in a solution of Zerotol (27.1% hydrogen peroxide and 2.0% peroxyacetic acid; Biosafe Systems, East Hartford, CT, USA) at a dilution of 1:100 with deionized water for 15 min and drained before transplanting.

### 4.2. Growth Conditions and Sole-Source Lighting Treatments

The experiment was performed in a temperature-controlled vertical farm on the ASU Polytechnic campus. Two vertical growing racks with three tiers were used to create six lighting treatments. Each tier had LED lamps on the top and a grow tray (1.12 m × 0.66 m × 0.18 m; GT24X44X7B; Botanicare, Vancouver, WA, USA) below. In each grow tray, 30 strawberry plants were transplanted into 36-cell floating rafts (122 cm × 61 cm × 2.5 cm; Beaver Plastics; Acheson, AB, Canada), excluding the outer guard rows. The distance between seedlings was 15 cm diagonally and 20 cm horizontally/vertically on each raft. Five weeks after transplanting, when each plant had reached canopy closure and its leaves touched the leaves of adjacent plants, 15 strawberry plants were removed in alternative rows to increase plant spacing. One 150-L water reservoir per growing rack supplied a nutrient solution made with tap water supplemented with a water-soluble fertilizer (15N-5P_2_O_5_-20K_2_O Tap Hydro FeED; JR Peters, Inc., Allentown, PA, USA). For the first two weeks after transplanting, the nutrient solution was made to provide the following nutrients (in mg∙L^−1^): 125 N, 18 P, 138 K, 25 Ca, 13 Mg, 1.25 Fe, 0.68 Mn, 0.43 Zn, 0.17 B, 0.17 Cu, and 0.008 Mo. After the two-week period following transplanting, nutrient concentrations increased by adding more fertilizer based on calculations to provide the following nutrients (in mg∙L^−1^): 150 N, 22 P, 166 K, 30 Ca, 15 Mg, 1.5 Fe, 0.8 Mn, 0.5 Zn, 2 B, 0.2 Cu, and 0.01 Mo. The nutrient solution was recirculated by a water pump (Eco 264; EcoPlus, Huntersville, NC, USA) and oxygenated with an air stone (728410; EcoPlus) and a 6.5-W air pump (Eco Air 4; EcoPlus). The pH was monitored daily with a pH and electrical conductivity (EC) meter (HI9814; Hanna Instruments, Woonsocket, RI, USA) and adjusted to 5.5–6.2 using potassium bicarbonate (Earthborn Elements, American Fork, UT, USA) or diluted (1:31) 95–98% sulfuric acid (J.Y. Baker, Inc., Phillipsburg, NJ, USA). The mean EC, pH, and water temperature (mean ± SD) of the nutrient solutions for two growing racks were (replication 1/replication 2): 1.82 ± 0.03/1.72 ± 0.04 mS∙cm^−1^, 5.91 ± 0.42/5.65 ± 0.79, and 20.3 ± 1.0/18.7 ± 0.8 °C, respectively, for the first two weeks after transplanting and 2.19 ± 0.17/2.37 ± 0.27 mS∙cm^−1^, 5.65 ± 0.30/6.07 ± 0.73, and 19.3 ± 0.9/19.9 ± 0.8 °C, respectively, for the remaining period.

The plants were grown under six different sole-source lighting treatments, which comprised combinations of two photoperiods (12-h and 16-h) and three PPFDs (200, 300, and 450 µmol∙m^−2^∙s^−1^) (Table 1). For three PPFD treatments, the PPFD of 200 µmol∙m^−2^∙s^−1^ was selected as a typical PPFD range used for indoor crop production [5], and higher PPFDs were created with 1.5-fold increases. Sole-source lighting treatments were provided by white + blue + red LED lamps (T8 Double-Row LED Indoor Grow Light; Homer Farms Inc., Mesa, AZ, USA) (Figure 6). The radiation spectrum of each treatment consisted of 35.2% blue (400–500 nm), 11.4% green (500–600 nm), 52.0% red (600–700 nm), and 1.5% far-red (700–800 nm) radiation. Four, six, and nine LED lamps were used to achieve the target PPFDs of 200, 300, and 450 µmol∙m^−2^∙s^−1^, respectively. The measurements of the PPFD for each lighting treatment were based on an average of 12 measurements from a spectroradiometer (PS-300; Apogee Instruments, Logan, UT, USA) made at predetermined horizontal positions at the surface level of the floating raft (21 cm below the LED lamps). Photoperiods of lighting treatments were controlled by programmable timers (GETA-US; CanaGrow, Shenzhen, Guangdong, China).

Plants were grown at an air temperature setpoint of 21 °C. At each tier of a growing rack, the air temperature was monitored and recorded once every 15 min by a thermometer hygrometer (H57075; Shenzhen Intellirocks Tech Co., Shenzhen, Guangdong, China). The average air temperatures (mean ± SD) during the experiment periods were (replication 1/replication 2): 20.6 ± 2.0/21.0 ± 1.3 °C, 21.2 ± 2.4/21.2 ± 1.6 °C, 20.7 ± 2.3/21.8 ± 1.9 °C, 20.5 ± 1.8/20.4 ± 1.2 °C, 20.7 ± 2.1/21.3 ± 1.7 °C, and 20.9 ± 2.2/21.1 ± 1.7 °C for the 12-h PPFD 200, 12-h PPFD 300, 12-h PPFD 450, 16-h PPFD 200, 16-h PPFD 300, and 16-h PPFD 450 treatments, respectively.

### 4.3. Data Collection and Analysis

For this study, strawberry plants were grown for 94 days (replication 1) or 115 days (replication 2) after transplanting. In each replication, 30 plants were transplanted per each lighting treatment and grown for five weeks. Five weeks after transplanting, for each lighting treatment, 15 plants were removed to increase plant spacing and 15 plants remained for each lighting treatment. At five weeks after transplanting, among the removed 15 plants, 10 random plants in each treatment were selected and the following data were collected on plants in each treatment: leaf number (when trifoliate leaves had unfolded), crown diameter, plant diameter, leaf length, leaf width, SPAD index (an indicator for chlorophyll concentration per unit leaf area), and the fresh and dry mass of the shoot and root. Crown diameter was measured at the base of plants between the highest roots and the plant canopy with a digital caliper (B07DFFYCXS; Adoric). Plant diameter was measured with a ruler between the two outermost contrary leaves. Leaf length or width were measured with a ruler in the largest leaf across the leaflets at the longest or widest part, respectively. SPAD index was measured at the central point of the leaflet between the midrib and the leaf margin of the largest leaf. Three readings per leaf were taken and averaged to a single SPAD index value for each plant. Fresh and dry mass (after plants were dried in an oven at ≥66 °C for ≥7 d) of the shoot and root were measured using an analytical balance (PB303-S; Mettler Toledo, Columbus, OH, USA).

Once each plant had its first open flower and first fully red fruit, the date of first flowering and date of first fruit harvest were recorded for each plant. Once plants had fully red fruits, fruits with a diameter ≥19.05 mm, the minimum diameter required for U.S. No.1. Grade strawberry fruits [44], were harvested, and fresh mass (using a scale (PB303-S; Mettler Toledo)) and fruit diameter were measured individually. Fruit diameter was measured at the widest point of the fruit by using a digital caliper (B07DFFYCXS; Adoric). The fruit harvest period was 30 days (replication 1) and 31 days (replication 2). In each replication, total fruit production (g∙m^−2^∙month^−1^) and total number of fruits (number∙m^−2^∙month^−1^) were calculated by dividing the total harvested fruit fresh mass per month (g∙month^−1^) and total number of fruits harvested per month (number∙month^−1^) for each lighting treatment by the growing area (0.74 m^2^).

The experiment used a randomized complete block design: each replication was regarded as a block; each tier of a growing rack was regarded as the experimental unit for the sole-source lighting treatment; and within each tier of a growing rack, each plant was the subsample or observational unit. Data were analyzed with SAS software (version 9.4; SAS Institute, Inc., Cary, NC, USA). Linear regression analysis for PPFD effects were conducted for each photoperiod with the PROC REG. Two-way analysis of variance was used to assess the effects of photoperiod at each PPFD using a PROC MIXED procedure (with two fixed factors for PPFD and photoperiod, two random factors of blocks (or replications) and interaction between blocks, PPFD, and photoperiod). For plant growth data, the data from individual plants (or observation units) were treated as single data points and totaled 120 (2 replications × 10 plants per replication per lighting treatment × 6 lighting treatments). For flowering and fruit production data (including days to flower, days to first fruit harvest, flowering % (or % of plants that flowered), fruiting % (or % of plants that produced fruits), total fruit production, total number of fruits, individual fruit fresh mass, and individual fruit diameter), the mean for each replication was treated as a single data point and included 12 data points (2 replications × 6 treatments).

## 5. Conclusions

In this study, we investigated the effects of PPFD, photoperiod, and DLI of sole-source lighting on plant growth, flowering, and fruit production in everbearing strawberry ‘Albion.’ During the vegetative growth stage, increasing the PPFD from 200 to 450 µmol∙m^−2^∙s^−1^ linearly increased shoot and root biomass and crown diameter, and increasing the photoperiod from 12 to 16 h increased root biomass. During the reproductive stage, extending the photoperiod from 12 to 16 h accelerated flowering (at a PPFD ≥ 300 µmol∙m^−2^∙s^−1^) and fruiting (at a PPFD of 450 µmol∙m^−2^∙s^−1^) and enhanced fruit production, but PPFD had little to no significant effects on flowering, fruiting, and fruit yield. Our results suggest that increasing the DLI by increasing the PPFD or photoperiod can increase strawberry plant growth, but extending the photoperiod can have a greater effect on promoting flowering and early fruit production than increasing the PPFD in strawberry ‘Albion.’

## Figures and Tables

**Figure 1 plants-12-00731-f001:**
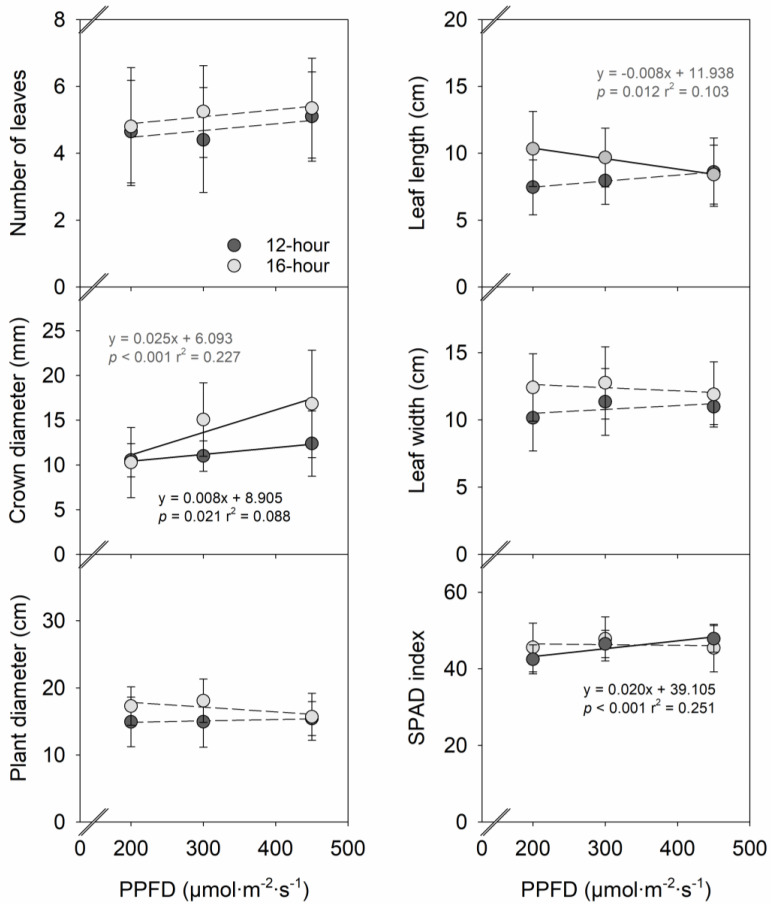
Influences of photosynthetic photon flux density (PPFD) of sole-source lighting treat-ments on plant growth characteristics of strawberry ‘Albion’ grown under a 12-h or 16-h photoperiod. Each data point represents the mean and standard deviation of two replications with 10 subsamples (plants) per replication (n = 20). Regression equations, coefficient of determination (r^2^), and *p* values are presented when the linear relationship with PPFD is statistically significant at *p* < 0.05 (solid line), but not when insignificant (dashed line).

**Figure 2 plants-12-00731-f002:**
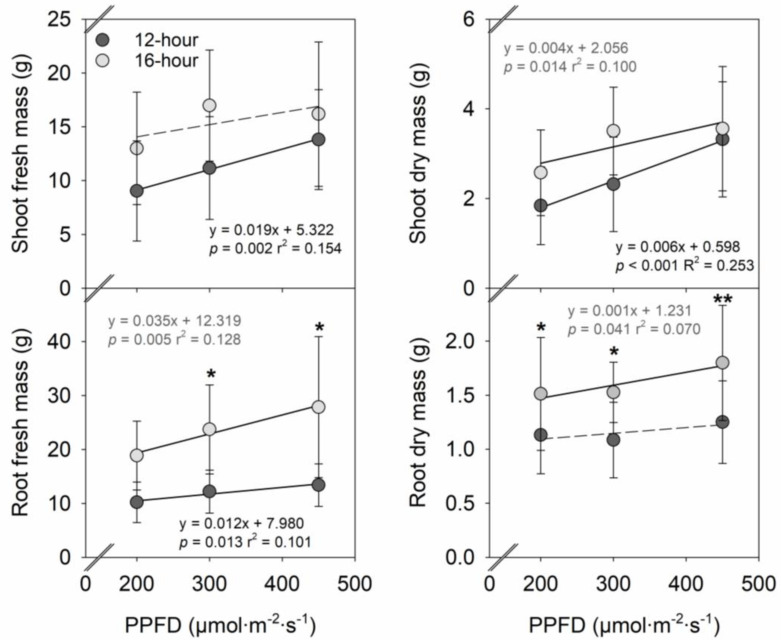
Influences of photosynthetic photon flux density (PPFD) of sole-source lighting treat-ments on fresh and dry mass of shoot and root of strawberry ‘Albion’ grown under a 12-h or 16-h photoperiod. Each data point represents the mean and standard deviation of two replications with 10 subsamples (plants) per replication (n = 20). Regression equations, coefficient of determination (r^2^), and *p* values are presented when the linear relationship with PPFD is statistically significant at *p* < 0.05 (solid line), but not when insignificant (dashed line). At any PPFD, * or ** indicates means at a 12-h and 16-h photoperiod are significantly different at *p* < 0.05 or 0.01, respectively.

**Figure 3 plants-12-00731-f003:**
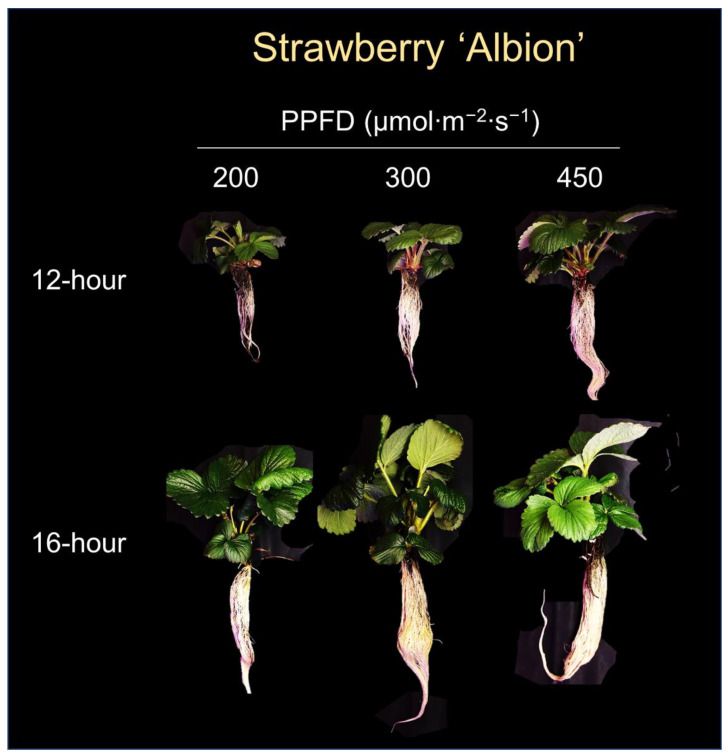
Strawberry ‘Albion’ plants grown for 5 weeks under six sole-source lighting treatments, which were comprised of combinations of two photoperiods (12-h and 16-h) and three photosynthetic photon flux densities (PPFDs) (200, 300, and 450 µmol∙m^−2^∙s^−1^). The photo was taken during the first replication.

**Figure 4 plants-12-00731-f004:**
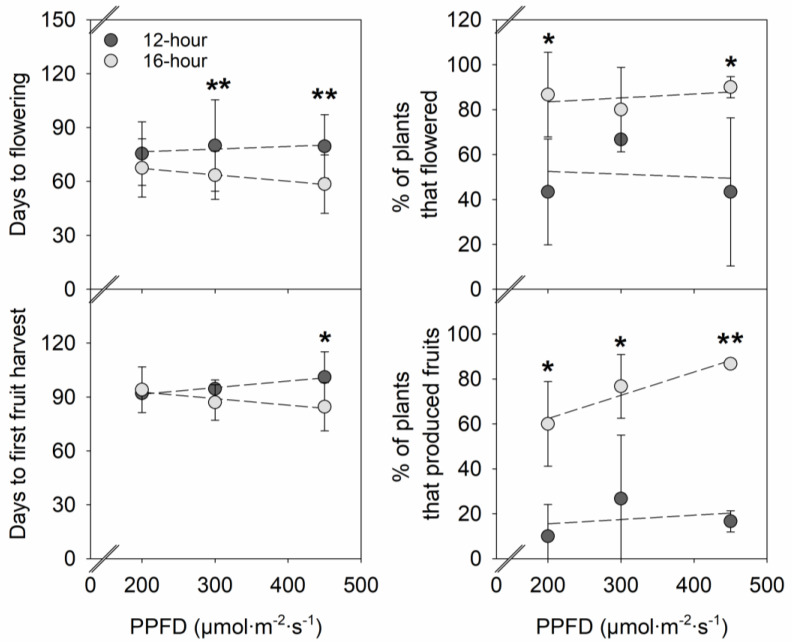
Influences of photosynthetic photon flux density (PPFD) and photoperiod of sole-source lighting treatments on days to first flower, days to first fruit harvest, percentage of plants that flowered, and percentage of plants that produced fruits of strawberry ‘Albion’. Each data point represents the mean and standard deviation of two replications (n = 2). At each photoperiod, linear relationship with PPFD is indicated with dashed line when insignificant (*p* ≥ 0.05). At any PPFD, * or ** indicates a significant difference at *p* < 0.05 or 0.01 at a 12-h or 16-h photoperiod, respectively.

**Figure 5 plants-12-00731-f005:**
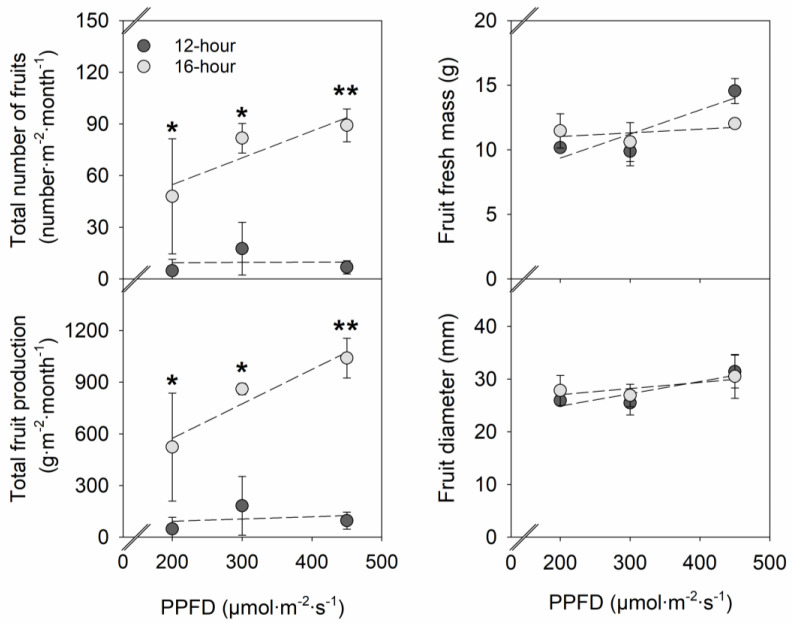
Influences of photosynthetic photon flux density (PPFD) and photoperiod of sole-source lighting treatments on total number of fruits, total fruit production, individual fruit fresh mass, and individual fruit diameter of strawberry ‘Albion’. Each data point represents the mean and standard deviation of two replications (n = 2). At each photoperiod, linear relationship with PPFD is indicated with dashed line when insignificant (*p* ≥ 0.05). At any PPFD, * or ** indicates a significant difference at *p* < 0.05 or 0.01 at a 12-h or 16-h photoperiod, respectively.

**Figure 6 plants-12-00731-f006:**
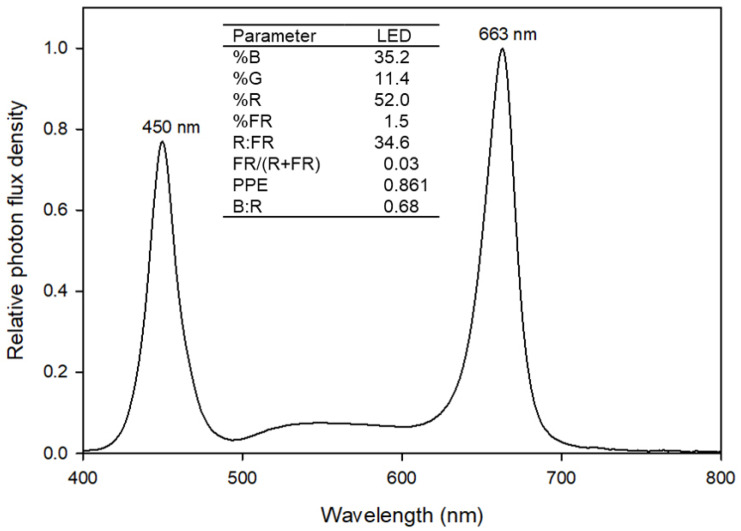
The spectral distribution of white + blue + red light-emitting diodes used for sole-source lighting treatments. The numbers above the peak of the spectral distribution indicate the peak wavelengths. The percentage (%) of blue (B, 400–500 nm), green (G, 500–600 nm), red (R, 600–700 nm), and far-red (FR, 700–800 nm) light were calculated as the ratio of the photon flux density integral of each waveband and total photon flux density (400–800 nm). R:FR was calculated as the ratio of the photon flux integral of R and FR light. FR fraction or ratio of photon flux integral of FR (700–800 nm) to the sum of the photon flux integral of R (600–700 nm) and FR was calculated following Kusuma and Bugbee [42]. Estimated phytochrome photoequilibria (PPE) was calculated following Sager et al. [43]. B:R was calculated as the ratio of the photon flux integral of B and R light.

**Table 1 plants-12-00731-t001:** Photoperiod (hour∙d^−1^), photosynthetic photon flux density (PPFD; µmol∙m^−2^∙s^−1^), and daily light integral (DLI; mol∙m^−2^∙d^−1^) of six sole-source lighting treatments.

**Sole-Source Lighting Treatments**		**Measured PPFD**	**Calculated DLI**
**Photoperiod**	**PPFD**		
12	200	207.4 ± 3.9	8.6
	300	310.5 ± 7.5	13.0
	450	441.3 ± 6.0	19.4
16	200	200.1 ± 8.7	11.5
	300	312.3 ± 10.2	17.3
	450	456.7 ± 4.8	25.9

## Data Availability

The data presented in this study will be openly available in FigShare at 10.6084/m9.figshare.21740273.

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
