# Peer review of "Growth, Flowering, and Fruit Production of Strawberry ‘Albion’ in Response to Photoperiod and Photosynthetic Photon Flux Density of Sole-Source Lighting"

_plants, 2023, doi:10.3390/plants12040731_

Round 1
Reviewer 1 Report
Line 44 : the sole-source lighting was used using LEDs in this study. Provide any references in which dual- or multi-source lighting are used.
line 77 : most studies ? (present appropriate references)
line 79 : few studies ? (present appropriate references)
Lines 77-80 : Present the main differences or originality of this study from the previous studies.
Line 87 : how many hypotheses are used in this study? Each research hypothesis should be tested using the experimental results.
Line 106 : R-squared represents a determination coefficient, NOT correlation one. Use a correct term!
Figures 1-3. each slope of each regression equation must be statistically tested using t-test. Use null hypothesis (slope is not statistically different from 0) against alternative hypothesis (slope is statistically different from 0).
Author Response
Thank you for your constructive suggestions and comments. Please see the attached.

Reviewer 2 Report
Yujin Park et al. investigated the effects of photosynthetic photon flux density (PPFD) and photoperiod of sole-source lighting on plant growth, flowering, and fruit production in strawberry ‘Albion.’ Their results are interesting but not sufficient for publishing at the present form. I suggest fruit quality should be investigated e.g. fruit size, color, taste and related components
Author Response

(The authors gave the same response as above.)

Round 2
Reviewer 2 Report
N/A